Natural Hazards of and Earth System Sciences

# Glacial lake change risk and management on the Chinese Nyaingentanglha in the past 40 years

#### Wang Shijin

State Key Laboratory of Cryospheric Sciences, Cold and Arid Regions Environmental and Engineering Research
 Institute, Chinese Academy of Sciences, Lanzhou, Gansu, 730000, China

7 \*Correspondence Address: Donggang West Road 320, Lanzhou, Gansu, China 730000.

- 8 Email: xiaohanjin@126.com.
- 9 Fax: +86 0931 4967336; Tel.: +86 0931 4967339.

#### 10

1

2 3 4

11 ABSTRACT. The paper analyzed synthetically spatial distribution and evolution status of 12 moraine-dammed lakes in the Nyainqentanglha Mountain, revealed risk degree of county-based potential 13 dangerous glacial lakes (PDGLs) outburst floods disaster by combining PDGLs outburst hazard, regional 14 exposure, vulnerability of exposed elements and adaptation capability and using the Analytic Hierarchy 15 Process and Weighted Comprehensive Method. The results indicate that 132 moraine-dammed lakes (>0.02 km<sup>2</sup>) with a total area of 38.235 km<sup>2</sup> were detected in the Nyainqentanglha in the 2010s, the lake number 16 17 decreased only by 5%, whereas total lake area expanded by 22.72%, in which 54 lakes with a total area of 18 17.53 km<sup>2</sup> are identified as PDGLs and total area increased by 144.31%, higher significantly than 4.06% of 19 non-PDGLs. The zones at very high and high integrated risk of glacial lakes outburst floods (GLOFs) 20 disaster are concentrated in the eastern Nyainqentanglha, whereas low and very low integrated risk zones 21 are located mainly in the western Nyainqentanglha. On the county scale, Nagque and Nyingchi have the 22 lowest hazard risk, Banbar has the highest hazard and vulnerability risk, Sog and Lhorong have the highest 23 exposure risk. In contrast, Biru and Jiali have the highest vulnerability risk, while Gongbo'gyamda and 24 Damxung have lowest adaptation capacity. The regionalization results for GLOF disaster risk in the study 25 are consistent with the distribution of historical disaster sites across the Nyainqentanglha.

KEYWORDS. moraine-dammed lake; potentially dangerous glacial lakes; disaster risks; assessment and
 regionalization; Nyainqentanglha Mountain

28

# 29 1. Introduction

The globally averaged combined land and ocean surface temperature data as calculated by a linear trend, shows a warming of 0.85 [0.65 to 1.06] °C, over the period 1880 to 2012. The total increase between the average of the 1850–1900 period and the 2003–2012 period is 0.78 [0.72 to 0.85] °C, based on the single longest dataset available (IPCC, 2013). Global warming has led to the rapid retreat of mountain glaciers, the formation of new glacial lakes, the expansion of existing glacial lakes (Yao, 2010) and increased potential for GLOFs (Clague and Evans, 2000; Nayar, 2009; Worni et al., 2012; Wang et al., 2015).

GLOF is low-frequency event, but it often causes enormous loss and damage of life, property
and human environment in downstream regions. GLOFs have frequently been reported in the
Himalaya, Peruvian Andes (Cordillera Blanca), Chilean Patagonia, Canadian Rockies,
Nyainq êntanglha range (Worni et al., 2013; Haeberli and others, 2013; Wang et al., 2015). For

41 example, over 21 GLOF disasters have been reported in Peru's Cordillera Blanca, killing 42 nearly 30, 000 people during the past 65 years (Carey, 2005, 2008; Carey et al., 2012). 43 According to past records, at least 16 GLOFs have occurred in Chinese Nyaing entanglha 44 since 1935 (Wang and Zhang, 2013). The occurrence of recent classic outbursts at a lake of 45 Palong Zangbu (i.e. river), 2007, Cilaco Lake of Nujiang River, 2009 and at Recireco lake of Yigong Zangbu, 2013 in Nyaing entanglha shows that the threat of GLOFs requires 46 appropriate and continued attention well into the 21st century, as glacier retreat continues 47 (Wang et al., 2015). Especially, on 5 July 2013, Recireco lake (an area of about  $57 \times 10^4$  m<sup>2</sup>) 48 49 outburst occurred in the eastern Nyaingêntanglha and then formed mudslides. As a result, 50 some persons were missing, numerous buildings were destroyed, and some infrastructures 51 were damaged. The total economic loss was estimated as 200 million RMB. Facts have 52 proved that: the economic losses caused by GLOF are much higher than the project costs to 53 early consolidate moraine dam and release flood waters.

GLOF disasters result from both natural and social factors and their interactions. GLOF risks 54 55 not only include the hazard of glacial lake outburst, but also involve the vulnerability and 56 adaptation capacity of exposed elements. The hazard of glacial lake outburst can be defined as 57 the product of outburst magnitude and outburst probability (Mckillop and Clague, 2007). However, GLOF impacts, regional exposure, the vulnerability of exposed elements and the 58 59 adaptation capacity downstream have received less consideration or synthetic and quantitative assessment in previous studies. In fact, glacial lake outbursts can be very difficult and 60 61 expensive to control, but regional exposure and the vulnerability of exposed elements 62 downstream can be reduced by improving adaptation capacity and risk management level (Wang et al., 2015). 63

This study analyzed synthetically county-based spatial distribution characteristics and 64 evolution status of moraine-dammed lakes and potential dangerous glacial lakes (PDGLs), 65 identified and analyzed GLOF disaster risk, and established a risk assessment system 66 including glacial lake outburst hazard, regional exposure, the vulnerability of exposed 67 68 elements and the adaptation capacity downstream. Finally, the study quantified the degree of risk of GLOFs in the study area using GIS technology, the analytic hierarchy process (AHP) 69 and the weighted comprehensive method (WCM). The study is not only of significance for 70 analyzing glacial lake outburst hazard and assessing the exposure and vulnerability of 71 72 exposed elements in Nyainqentanglha, but also has important theoretical reference to provide 73 a scientific support for prevention and mitigation planning of GLOF disaster, infrastructure 74 construction, industry distribution and village land use planning in GLOF affected areas.

## 75 2. Study area

76 Nyainqentanglha range, with the leghth of about 740 km, locates in the border belt between 77 the Indian and Eurasian plate. The highest peak located in the mostwestern part of Nyainq ântanglha with the altitude of 7 162 m (Fig. 1). The study region  $(90.50^{\circ} - 97.80^{\circ} \text{E})$ ; 78 29.15 °- 32.20 °N) covers an area of 48.82 km<sup>2</sup>, accounting for 39.74% of the land area of the 79 Tibet Autonomous Region. It is bordered to the south by Yarlung Zangbo (Brahmaputra 80 River), to the west by Qiongmugang peak (7 048 m) on North of Ma River, to the north by 81 Tanggula Mountain, and to the east by Hengduan Mountain (Fig. 1). Nyainqentanglha 82 83 developes a large number of modern glaciers and is one of largest glaciation center of the 84 middle and low latitude in the world. The second Glacier Inventory (Liu et al., 2015) shows

Nyainq êntanglha exists 6 860 glaciers with an area of 9 559.20 km<sup>2</sup> in 2010s and the number and area of glaciers decreased by 3.11% and 10.67% respectively in nearly 40 years.

87 Nyaingentanglha is divided into eastern and western sections in the headwater of Lhasa River (Wu et al., 2002). The eastern Nyaingentanglha is controlled by the Southwest Indian Ocean 88 89 monsoon with warm-humid climate. Because warm-humid climate is forced by the steep terrain to uplift, here becomes one of most rainfall area and the wettest region in the Tibetan 90 91 Plateau. The precipitation can reach to 3 000 mm around in some glacier areas (Jiao et al., 2005) and glacier area basically accounts for more than 90% of total glacier area in 92 93 Nyainqentanglha. Here, neotectonics is strong, and tectonic seismic activity is also frequent and intense. The earthquake often destroyed the stability of glacier body and moraine dam 94 95 and made own state of glacial lake lose balance. And, it also damaged geotechnical stability, 96 made it generate adequate loose materials, which provides material source condition for 97 GLOF mudslides.

98

Fig. 1 Location of the Nyainqentanglha Mountain showing the major river basins, glacial lake
 distribution, recorded GLOF sites and 11 county boundaries in Southeastern Tibetan Plateau

101 The study area is administratively divided into 11 counties: Sog, Nagqu, Damxung, Lhorong, 102 Nyingchi, Jiali, Gongbo'gyamda, Bomi, Banbar, Biru, Basu County (Fig. 1). Most 103 communities located in remote areas away from glacial lakes are often within the reach of 104 GLOFs. In 2013, the population and regional GDP were 550, 000 and RMB 3.63 billion, respectively, accounting for 17.63% and 4.44% of total population and GDP of the Tibet 105 Autonomous Region. Local residents' perception of the threat from remote glacial lake is 106 107 relatively weak, and their ability to prevent and adapt to disaster is extremely limited, which provides a disaster-affected body conditions for the formation of GLOF disaster. It can be 108 predicted that GLOF impacts will likely extend farther downstream as glaciers continue to 109 110 retreat in the next few decades.

## 111 **3.Data and Processing**

## 112 **3.1. Date**

113 The data for this study consist of Landsat imagery obtained on multiple dates in the 1990s and

114 2010s, topographic maps, ASTER DEMs in 2009, and statistical data concerning the socio-economic system in 2014. Image data are used to analyze the spatial and temporal 115 116 variation of glacial lakes and to identify their potential risk, whereas regional socio-economic data are used to assess exposure, vulnerability and adaptation capacity in hazard-affected 117 118 regions. The Nyainqentanglha, especially, the middle-eastern region, is affected by the southwes monsoon with humid and heat climate, thus the region is covered with cloud during 119 most of month in a year. To avoid cloud and snow cover during the monsoon and ensure 120 121 minimal snow coverage, we selected as far as possible 22 satellite images as far as possible with 

156

$$U_{A} = \frac{(2U_{L})}{\sqrt{\sum \lambda^{2}}} \times \sum \lambda^{2} \times \sum \sigma^{2}$$
<sup>(2)</sup>

where  $U_L$  is the linear uncertainty (m),  $U_A$  is the glacial lake's area uncertainty (m<sup>2</sup>),  $\lambda$  is the 157 original pixel resolution of each individual image (m) and  $\sigma$  is the co-registration error of 158 each individual image to topographic maps (m). Accordingly, the maximum error in 159 160 coregistration  $(U_A)$  for changes in glacial lakes from the 1990s to the 2010s was calculated as 161  $\pm 0.015 \text{ km}^2$ .

#### 162 3.3 Identification of potentially dangerous glacial lakes

Generally, glacial lake failure comes from external (ice/snow/rock avalanches, landslides, 163 164 rainstorm, glacier advance, earthquake, snow and ice melting) and internal incentives (the ablation of buried ice within moraine dam, the release of lake water inside ice body, piping, 165 seepage, etc). According to failure mechanisms, GLOF can be divided into five categories: 1) 166 The outburst flood is triggered by wave overtopping moraine-dam, 2) by seepage/piping; 3) 167 by flow water erosion from next valley; 4) by earthquake; 5) by various combined factors. 168

Based on above analysis and taking into account the availability of data obtained, this study 169 170 selected only moraine-dammed lake area (>  $0.02 \text{ km}^2$ ), the rate of lake area increase (> 20%), the distance between lake and glacier snout (