# Peer review of "Glacial lake change risk and management on the Chinese Nyaingentanglha in the past 40 years"

_Natural Hazards and Earth System Sciences, 2016_

## Referee Comment (RC1) · Anonymous Referee #1 · 24 Oct 2016

The authors present a quantitative study on glacier lake outburst flood (GLOF) risk in a mountain range in the southern Tibetan Plateau. The hazard assessment is based on remote sensing data from two time periods. Risk quantification uses weighted socio-economic index data on exposure, vulnerability and adaptive capacity. The manuscript represents a one-to-one adaptation of a study previously performed in the Chinese part of the Himalayas, published in 2015 (Wang et al. 2015; Journal of Glaciology). Though the scientific work seems to be acceptable, the study does not provide new concepts or new ideas, but represents a case study of the previously established method. However, the manuscript is presented in a poor fashion which does not make up to the quality of standards of NHESS. The English language is poor in many parts and contributes to some confusion while reading, for example in the application of the terms hazard, risk, vulnerability and exposure. This is crucial for a publication focussing on risk. Other

more formal issues concern the presentation of numbers in the text, the use of units and the composition of figures and tables. In more detail, the manuscript lacks to thoroughly explain how index parameters for the quantification of risk are weighted. Finally, the issue in risk management, as included in the title, is not given much attention. The comments on management in the discussion/conclusion are not specific or innovative, but rather read as general management issues as found in textbooks on disaster and risks. To conclude, the study presents a risk assessment for a specific hazard process and a specific region. The method is not new and the conclusions drawn from this study are not very specific, except for the information that the region is "another high-frequency and severely affected area of GLOF disaster" an information that has not been generated by this study. I therefore reject the manuscript.

---

## Author Comment (AC1) · 25 Oct 2016

The manuscript used a study method previously performed (Wang et al. 2015; Journal of Glaciology). The aim is to promote this integrated risk assessment method. GLOF disasters result from both natural and social factors and their interactions. GLOF risks not only include the hazard of glacial lake outburst, but also involve the vulnerability and adaptation capacity of exposed elements. Previous GLOF studies have rarely considered social factors. In order to limit the number of words, how to determine the weights of 15 indicator factors are ignored, but the basic steps or process are provided. Likewise, in order to limit the number of words,the study put forward a general risk management method, but not specific measures. If the number of words allowed, the study can provide specific risk management measures of GLOF disaster into the text. The following shows only the headings of the specific management measures to adapt to GLOF

disaster. Specific measures are as follows:1) Monitor regularly glacial lake dynamics and examine and check mainly dangerous glacial lakes; 2)Take reasonable engineering measures and control effectively risk from dangerous glacial lakes;3)Implement many-sided participatory mechanisms and enhance integrated disaster prevention and mitigation capabilities;4)Carry community-based risk management mechanism and improve mass-based monitoring and prevention system;5)Implement disaster assessment planning and strengthen preparedness capability of disaster;6)Take advantage of advantages and avoid disadvantages, use efficiently water conservancy and hydropower resources of glacial lake. The author believes that the promotion of this risk assessment and management methods have a certain theoretical and practical significance.

---

## Referee Comment (RC2) · Anonymous Referee #2 · 21 Nov 2016

In this article, Wang analyzed the spatial distribution and evolution of glacial lakes in the Nyainqentanglha Mountain, identified potential dangerous glacial lakes, and assessed the GLOF disaster risk. My suggestion is that a significant revision is needed before it is accepted. Some main comments are as the following: First, the author has published similar paper (Journal of Glaciology, 2015) using similar method with an exception of study area. The previous paper studied the glacial lakes in Himalaya region and this paper studied Nyainqentanglha Mountain. Could the author explain what is new in the paper, including new study method, or new findings. Second, as the author has analyzed the spatial pattern and evolution of glacial lakes in Himalaya region, some similarity and difference between Himalaya and Nyainqentanglha Mountain should be compared and discussed. Thirdly, the author mentioned the method of uncertainty estimation, but it is not shown in the result. Fourthly, there are many grammatical

mistakes in the paper. Substantial language improvement is needed.

Specific comments L35 Abbreviation of "GLOFs" should be given the full name when it appears first time in the main text though you show in the abstract. L78-79 The author obviously gives a wrong number of the area of study region. L104-106 For the population and GDP, the author should give the source of the data since this is very important for the exposure and vulnerability analysis. L119 "southwest" should be "southwest" L121 "as far as possible" repeated twice, delete one of them L162-177 For this section of identification of PDGLs, I cannot follow the author's rationale. For example, why you select the four criteria to identify the PDGLs? This section should be significantly improved. Figure 4 For the degrees of hazard, exposure, vulnerability and adaptation capacity, how you classify as very low to very high?

Please also note the supplement to this comment:
http://www.nat-hazards-earth-syst-sci-discuss.net/nhess-2016-300/nhess-2016-300-RC2-supplement.pdf